# Two casting methods compared in patients with Colles' fracture: A pragmatic, randomized controlled trial

**Lauri Raittio**[1]*, **Antti P. Launonen**[2], **Teemu Hevonkorpi**[1], **Toni Luokkala**[3], **Juha Kukkonen**[4], **Aleksi Reito**[2], **Minna K. Laitinen**[5,6], **Ville M. Mattila**[1,2]

**1** Faculty of Medicine and Health Technology, Tampere University, Tampere, Finland, **2** Department of Orthopaedics, Unit of Musculoskeletal Surgery, Tampere University Hospital, Tampere, Finland, **3** Central Finland Central Hospital, Jyväskylä, Finland, **4** Satakunta Central Hospital, Pori, Finland, **5** Division of Orthopaedics and Traumatology, Unit of Musculoskeletal Surgery, Tampere University Hospital, Tampere, Finland, **6** Department of Orthopaedics and Traumatology, Helsinki University Central Hospital, Helsinki, Finland

* lauri.raittio@tuni.fi

## Abstract

### Background

Distal radius fractures are common fractures and the cornerstone of treatment remains immobilization of the wrist in a cast. At present, there is a scarcity of studies that compare different cast immobilization methods. The objective of the study was therefore to compare volar-flexion and ulnar deviation cast to functional cast position in the treatment of dorsally displaced distal radius fracture among elderly patients.

### Methods and findings

We performed a pragmatic, randomized, controlled trial in three emergency centers in Finland. After closed reduction of the fracture, the wrist was placed in either volar-flexion and ulnar deviation cast or functional cast position. The follow-up was 12 months. The primary outcome was patient-rated wrist evaluation (PRWE) score at 12 months. The secondary outcomes were Quick-DASH score, grip strength, health-related quality of life (15D), and pain catastrophizing scale. The number of complications was also recorded. In total, 105 participants were included in the study. Of these, 88% were female and the mean age was 73.5 (range 65–94) years. In the primary analysis, the mean difference in patient-rated wrist evaluation measure between groups was -4.9 (95% CI: -13.1.– 3.4., p = .24) in favor of the functional cast position. Operative treatment due to loss of reduction of fracture was performed for four patients (8%) in the FC group and for seven patients (13%) in the volar-flexion and ulnar deviation cast group (OR: 0.63, 95% CI: 0.16–2.1).

### Conclusion

In this study, the data were consistent with a wide range of treatment effects when comparing two different cast positions in the treatment of distal radius fracture among elderly

**Data Availability Statement:** All relevant data are within the manuscript and its Supporting Information files.

**Funding:** The Academy of Finland https://www.aka.fi/en/ (grant number: MS773, VM) did grant the funding based on the research protocol for the overall maintenance of the trial. The funders had no role in study design, data collection and analysis, decision to publish, or preparation of the manuscript.

**Competing interests:** The authors have declared that no competing interests exist.

**Abbreviations:** CR, closed reduction; DRF, distal radius fracture; ENMG, electroneuromyography; FC, functional cast; qDASH, Quick-DASH; PCS, Pain Catastrophizing Score; PROM, patient-reported outcome measure; PRWE, Patient-Rated Wrist Evaluation; VFUDC, volar-flexion and ulnar deviation cast.

patients at 12-month follow-up. However, the functional cast is more likely to be superior when compared to volar-flexion and ulnar deviation cast.

## Trial registration

ClinicalTrials.gov identifier: NCT02894983 Accessible: https://clinicaltrials.gov/ct2/show/NCT02894983

## Introduction

Distal radius fracture (DRF) is the most common fracture in adults, and patients aged over 65 years are most at risk of suffering DRF [1]. Over the years, various immobilization methods have been described for the treatment of DRFs [2–6]. There is, however, a scarcity of studies that compare different cast immobilization methods and there is not enough good-quality evidence to enable the selection of one preferable method over the others [7].

The volar-flexion and ulnar deviation cast (VFUDC) immobilization method was first described by Cotton in 1910 [8]. The VFUDC is still widely used, since it is thought to maintain the fracture position with ligamentotaxis [9]. In another common immobilization technique, functional cast (FC) position, the wrist is stabilized in 0–20 degrees of dorsal angulation, supposedly allowing better ability to function to be maintained and rehabilitation to the wrist and hand.

Radiographs have been used by researchers and physicians to identify and classify DRFs, to select treatment interventions, and to predict the prognosis of patients with DRF. However, the association between radiographic parameters and the prognosis of DRF measured with patient-reported outcome measures (PROMs) is obscured with the dorsally displaced DRFs of patients aged 65 or more [10–16]. More recently, various psychological factors, such as depression, catastrophic thinking related to pain, and anxiety, have been shown to be associated with a decrease in the outcomes of DRF treatment [17–19].

In this pragmatic [20–22], randomized, controlled, multicenter trial, we compare the two commonly used cast immobilization positions, VFUDC position and FC position (Figures in S4 Appendix), in primarily successfully reduced, dorsally displaced DRFs in patients aged 65 or more. It was hypothesized that VFUDC and FC positions result in similar functional results. Furthermore, we hypothesized that catastrophic thinking related to pain would worsen our primary PROM results, patient-rated wrist evaluation (PRWE) score, regardless of the treatment group at 12 months.

## Methods

### Trial design

This study is a pragmatic, randomized, controlled multicenter trial that compares two cast immobilization positions, VFUDC position and FC position, of dorsally displaced DRFs in patients aged 65 years and more. The study was conducted at 3 large emergency hospitals in Finland; Tampere University Hospital (Tampere), Central Finland Central Hospital (Jyväskylä), and Satakunta Central Hospital (Pori). The study protocol has been published previously [23]. Ethical approval was obtained from each study center. The authors have written the manuscript together and made the joint decision to submit the manuscript for publication. There were a few changes made in the methods and design after the protocol was published. These changes have been identified and described in more detail in appendix (S2 Appendix). The trial was registered two

months after the first recruitment in the study by the corresponding author after the summer holiday season. A total of three patients were recruited before the registration of the trial at Tampere University Hospital, the main study center. The authors confirm that all ongoing and related trials for this intervention have been registered. The Regional Ethics Committee of Tampere University Hospital approved the protocol of the trial and additional papers, including consent form, patient information sheet, and questionnaires 4/1/2016 (Approval number: ETL R16035). The patients were recruited between July 2016 and May 2017 and followed until May 2019.

## Enrolment and randomization

All consecutive patients aged 65 years and more with a successfully reduced, dorsally displaced DRF were eligible for inclusion (criteria are presented in S1 Appendix). In this pragmatic study, there were no definitive radiographic criteria for the inclusion, but the national care guidelines followed by the recruiting on-call physicians set the parameters for successfully reduced, dorsally displaced DRF as follows: dorsal angulation < 15 degrees, radial shortening < 3 mm, volar angulation < 20 degrees, intra-articular step < 1 mm, and radio-ulnar angulation < 15 degrees [24]. All the patients were able to ask for additional information about the trial and informed consent was obtained from all enrolled patients.

After the diagnosis and assessment of informed consent, the patients were randomized to either VFUDC or FC using a random number matrix in block allocation in 1 to 1 ratio fashion. The blocks were stratified by age (65 to 74, and 75 or older), and intra-vs. extra-articular fracture [25,26]. The treatment allocations from the random number matrix were situated in the emergency room in sealed envelopes.

## Procedure

Closed reduction of the fracture was performed under local anesthesia and then the wrist was placed in either functional or VFUD position. Both types of plaster cast were in the form of below-elbow cast. Each site had an example-cast of both immobilization methods as a gold-standard of the final position.

## Outcome measures

The primary outcome measure was the mean difference in Patient-Reported Wrist Evaluation (PRWE) score between study groups at 12 months. The PRWE has 15 questions regarding the subjective ability to function of the wrist and hand rated on an 11-point scale from 0 to 10, giving a total range of 0 to 100 (in which 0 is the best). Secondary outcome measures were the short version of Disabilities of arm, shoulder and hand, Quick-DASH (qDASH, in a scale of 0–100) score, visual analogue scale of pain (VAS, in a scale of 0–100), health-related quality of life (15D, in a scale of 0–1), grip strength, number of complications, number of surgical interventions, number of cast changes, and radiographic parameters. The radiographic parameters were assessed by one of the authors who was blinded to the treatment group at the time of assessment. Catastrophic thinking related to pain was assessed using the pain catastrophizing scale (PCS). The questionnaires with unanswered responses were analyzed by the standards of the user's manual of each patient-reported outcome measure.

## Statistical analysis

A minimum of 40 patients per group was needed to detect the minimal clinically important difference (MCID) of 11 points on the PRWE scale with standard deviation of 14 points [27], a power of 80%, and at a 0.05 confidence level. The drop-out rate was estimated to be 30%.

The primary analysis included all patients who completed the questionnaires at 12-month follow-up. The secondary analysis was performed by including patient loss at follow-up by multiple imputation method. Differences between the two treatment groups in PRWE score, qDASH score, VAS, 15D, grip strength, effect of cast changes on PRWE, number of complications, internal validity of inclusion by post-reduction radiographs, and variance of continuous variables were analyzed using the Student's t-test. Two-way tables with the Fisher exact test were used for dichotomous variables for p-values and logistic regression for confidence intervals. The effect of cast treatment on the number of cast changes was analyzed using the Pearson's chi-square test. The 95% confidence intervals (CI) were calculated for the differences between the selected outcome measures. To assess the linear correlation between two continuous outcome measures, Pearson's correlation coefficients (CC) were calculated. In a subgroup analysis, the effect of various baseline characteristics were analyzed against the PRWE score at three and 12 months in multivariate linear regression. SPSS statistical software, version 25, was used for the analyses in the study. Several secondary analyses were performed in addition to the analyses presented in the protocol. These changes have been identified and described in the appendix file (S3 Appendix).

## Follow-up

One to 2 weeks after casting, a control appointment was arranged to evaluate the fracture position. If the reduction in fracture alignment was lost, operative treatment was considered by surgeons in accordance with the preferences of the patients in a pragmatic manner. Otherwise patients underwent a five-week cast immobilization period. Guided physiotherapy was introduced if needed (Table 1)

Patients were analyzed according to the intention-to-treat principle. The follow-up was organized in each center up to three months, and thereafter questionnaires were collected at Tampere University Hospital at 12 months. The assessment table is provided in the protocol of the study [23].

## Results

### Participants

From July 2016 to May 2017, a total of 105 patients from 3 Finnish hospitals were recruited and randomized into the trial and followed up for 12 months until May 2018. The baseline characteristics of the patients are shown in Table 1. The mean age at the time of recruitment was 73.5 years, and 88% of patients were female.

A total of 55 patients were assigned to the FC group and 50 patients to the VFUDC group. Thirteen patients withdrew from the study before the three-month clinical visit and 19 patients

**Table 1. The baseline characteristics of patients in the and VFUDC and FC groups.**

| Characteristic | FC | VFUDC |
|---|---|---|
| Randomized patients | 50 | 55 |
| Age, mean (range) | 74.6 (65–94) | 72.6 (65–89) |
| Sex (female/male) | 44/6 | 48/7 |
| Use of ancillary outside or inside of home (yes/no) | 10/34 | 3/47 |
| Dominant hand (right/left) | 47/2 | 50/5 |
| Fracture side (right/left) | 18/32 | 27/28 |
| Distance span in week (walk or cycle) in km, mean (SD) | 3.5 (3.2) | 3.3 (2.9) |
| Pain catastrophizing scale, mean (SD) | 9.4 (11.1) | 9.3 (12.1) |
| Extra-/Intra-articular fractures | 32/18 | 42/13 |
| Use of guided physiotherapy after the fracture, yes (%) | 21 (48%) | 24 (48%) |

did not return the questionnaires mailed to them at 12 months after recruitment. Therefore, 86 patients were included in the primary, intention-to-treat analysis at 12 months (Fig 1).

### Outcomes at 3 and 12 months

**PROMs and health-related quality of life.** The primary outcome measure, PRWE score with intention-to-treat analysis, was measured at 12 months. Results from the primary and secondary outcome measures are summarized in Table 2. At 12 months, the mean (CI for difference in means) PRWE score was 15.5 and 20.4 (-13.1–3.4, p = .24), the qDASH score was 17.2 and 20 (-10.4–5.0, p = .47), and the VAS was 12.6 and 15.6 (-10.9–4.9, p = .51) for the FC and

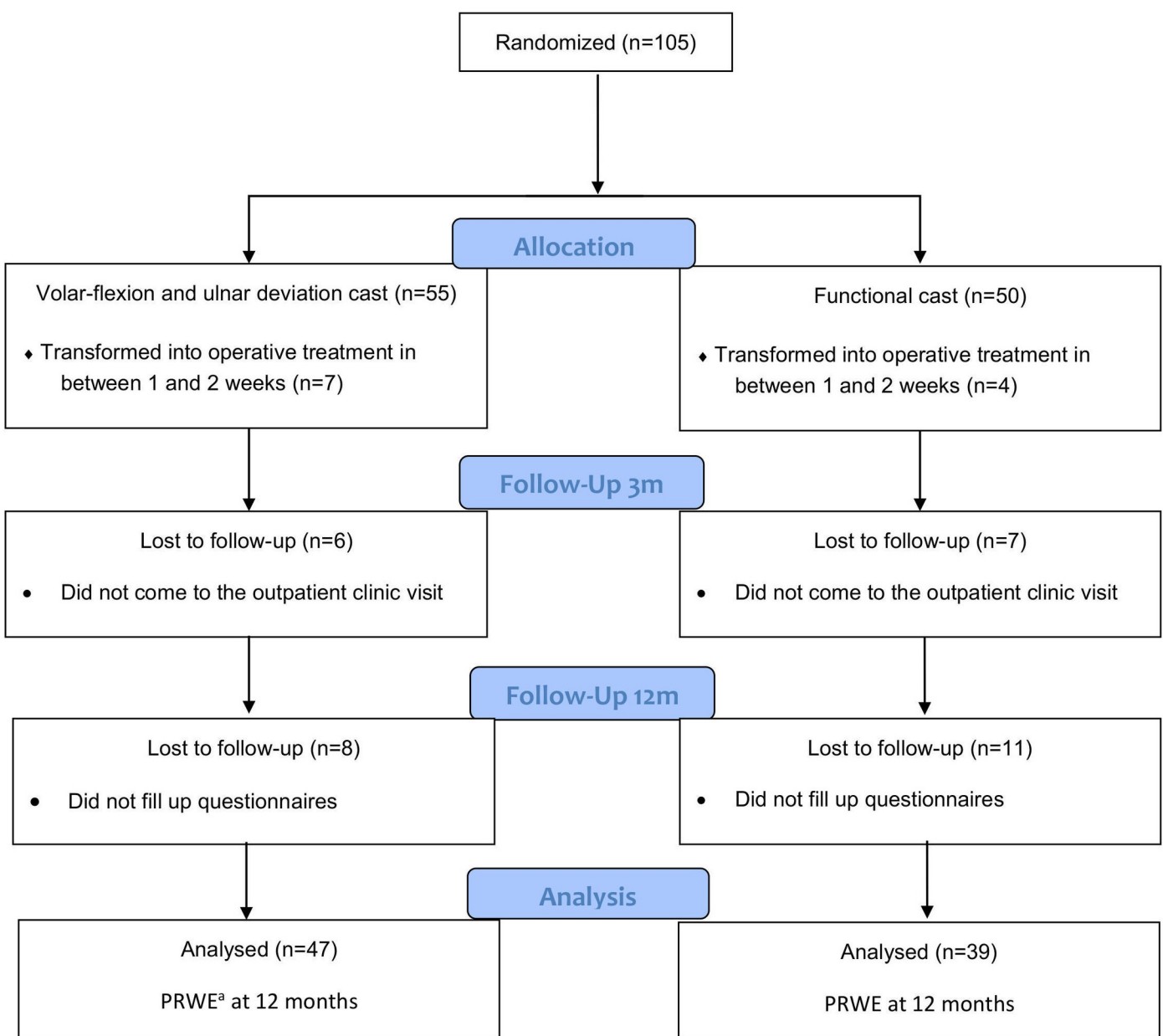

**Fig 1. The flow diagram of the study (CONSORT).** [a]PRWE = patient-rated wrist evaluation.

**Table 2. Primary, intention-to-treat analysis of outcome measures for the VFUDC and FC treatment groups at baseline, three, and twelve months after dorsally displaced DRF.**

| Evaluation | VFUDC group | N | FC group | N | SD total | Difference between groups (95% confidence interval) | P-value |
|---|---|---|---|---|---|---|---|
| | | | | | | | T-test |
| PRWE | | | | | | | |
| 3 months | 36.0 | 47 | 30.6 | 44 | 22.4 | -5.4 (-14.7–3.9) | .25 |
| 12 months | 20.4 | 47 | 15.5 | 39 | 19.2 | -4.9 (-13.1–3.4) | .24 |
| Quick-DASH | | | | | | | |
| 3 months | 34.7 | 44 | 31.3 | 38 | 22.9 | -3.4 (-13.5–6.7) | .51 |
| 12 months | 20.0 | 46 | 17.2 | 38 | 17.5 | -2.8 (-10.4–4.9) | .47 |
| 15D | | | | | | | |
| 3 months | 0.89 | 46 | 0.87 | 43 | 0.12 | -0.02 (-0.07–0.03) | .40 |
| 12 months | 0.89 | 47 | 0.87 | 39 | 0.11 | -0.01 (-0.06–0.04) | .61 |
| VAS (mm) | | | | | | | |
| 3 months | 24.3 | 50 | 21.0 | 44 | 19.6 | -3.3 (-11.4–4.7) | .41 |
| 12 months | 15.6 | 46 | 12.6 | 38 | 18.1 | -3.0 (-10.9–4.9) | .51 |
| PCS | | | | | | | |
| Baseline | 9.3 | 52 | 9.4 | 42 | 11.6 | 0.1 (-4.6–4.8) | .97 |
| 3 months | 10.0 | 45 | 8.8 | 40 | 10.9 | 1.2 (-3.6–5.9) | .71 |
| 12 months | 12.2 | 38 | 8.6 | 35 | 13.0 | -3.6 (-9.6–2.5) | .38 |

VFUDC groups, respectively. The mean (CI) grip strength of the fractured side in proportion to the controlled side measured at 3 months was 51% for the FC group and 45% for the VFUDC group (-0.26–0.14, p = .17). The changes during follow-up and differences between study groups in 15D were nonexistent in the study (Table 2). The correlation between the grip strength of the uninjured limb at 3 months did not correlate with the PRWE at 3 months (CC: 0.01, p = .92). The correlation between 15D and PRWE was present at 3 months (CC: -0.41, p < .001) and at 12 months (CC: -0.41, p < .001).

## Radiographs

The measured indicators from radiographs showed a small difference between the groups (Table 3). The radiographic confirmation of the differences between the two immobilization methods was performed using measurements of ulnar deviation of the third metacarpal compared with the radial axis and angulation of the wrist comparing second metacarpal flexion/extension to radial axis, having differences in means (CI) -4.0 (-6.0 - -2.0, p < .001) and 12.4 (8.2–16.2, p < .001) degrees of difference between the groups, respectively.

## PCS

The PCS score at the time of recruitment did not correlate with PRWE score, qDASH score, or VAS at 12 months, although the PCS score at 3 months did correlate moderately with the PRWE score (CC: 0.19, p = .1), and slightly with the qDASH score (CC:0.14, p. = .24), and VAS (CC: .17, p = .17) at 12 months. An increase in the patient-specific PCS value between baseline and 12 months did correlate moderately with a worsening (increase in value) of the PRWE score at 12 months (CC: 0.24, p = .05).

## Complications

Complications are summarized in Table 4. Operative treatment due to loss of reduction of fracture was performed for four patients in the FC group and for seven patients in the VFUDC

**Table 3. Mean values of radiographic outcome measures and mean differences between VFUDC and FC position groups.**

| Evaluation | VFUDC group | N | FC group | N | Difference between groups (95% confidence interval) | P-value T-test |
|---|---|---|---|---|---|---|
| Ulnar variance | | | | | | |
| Before CR* | 3.7 | 55 | 3.4 | 50 | -0.3 (-1.5–0.6) | .59 |
| After CR | 1.1 | 55 | 1.5 | 50 | 0.4 (-0.4–1.2) | .69 |
| At 3 months | 3.2 | 50 | 4.4 | 43 | 1.2 (0.1–2.4) | .04 |
| Inclination | | | | | | |
| Before CR | 15.5 | 55 | 17.0 | 50 | 1.5 (-0.2–3.2) | .09 |
| After CR | 18.8 | 55 | 18.8 | 50 | -0.02 (-1.4–1.4) | .98 |
| At 3 months | 17.2 | 50 | 16.3 | 43 | -0.9 (-2.6–0.8) | .30 |
| Dorsal angulation | | | | | | |
| Before CR | 27.5 | 55 | 23.7 | 50 | -3.8 (-8.1 –-0.5) | .09 |
| After CR | 7.4 | 55 | 8.8 | 50 | 1.4 (-1.6–4.4) | .37 |
| At 3 months | 12.8 | 50 | 11.9 | 43 | -0.9 (-5.2–3.4) | .67 |
| Extra- / Intra-articular (% extra) | 42/13 (76%) | | 32/18 (64%) | | | .20 |
| 3-MCP-Radius ulnar deviation | | | | | | |
| After CR | 8.6 | | 4.5 | | -4.0 (-6.0 –-2.01) | .00 |
| Wrist flexion (+) or extension (-) | | | | | | |
| After CR | 15.5 | | 3.1 | | -12.4 (-16.3 –-8.4) | .00 |

*CR = closed reduction.

group. Pain and stiffness were reported at the outpatient clinic at 3 months, and both pain and stiffness were found to be less in the FC group (Table 4). The number of cast changes was reported by the patients. During the five-week cast immobilization period, 14 cast changes occurred in the FC group compared with 25 cast changes in the VFUDC group (p = .55). In addition, the mean difference in PRWE score at 12 months was -8.5 points (CI: -18.0–1.1, p = .08) lower if no cast changes occurred compared with one or more cast changes. In the VFUDC group, two patients had electroneuromyography (ENMG), compared to none in the

**Table 4. Reported complications, operative treatment, and cast changes among study groups.**

| Evaluation | VFUDC | FC | Odds ratio* (95% CI) | P-value Fisher exact |
|---|---|---|---|---|
| Outpatient visit at 3 months, yes/no (%) | | | | |
| Reported pain | 14/39 (26%) | 4/42 (9%) | (0.08–0.90) | .04 |
| Reported stiffness | 5/48 (9%) | 0/46 (0%) | (0.00 - ) | .06 |
| Questionnaires at 12 months, yes (%) | | | | |
| Operative treatment | 7/55 (13%) | 4/50 (8%) | (0.16–2.13) | .45 |
| Questionnaires at 12 months | | | | |
| Cast changes | | | | |
| 0 | 35 | 34 | | |
| 1 | 7 | 6 | | |
| 2 | 6 | 4 | | |
| 3 | 2 | 0 | | .55** |

*The logistic regression was performed to calculate confidence intervals for odds ratio.

**The Chi square test was utilized to calculate the p-value.

FC group, performed for the injured wrist and hand during the follow-up period due to complaints of numbness, but none needed surgery for the condition.

## Secondary analysis and linear regression

After adjustment by patients lost to follow-up using the multiple imputation method, between group differences in means (CI) at 12 months for PRWE score was -3.9 (-12.0–4.2, p = .34), for qDASH score -2.1 (-9.4–5.3, p = .58), and for VAS at 12 months -2.5 points (-10.0–4.9, p = .50) in favor of the FC group.

In linear regression, variance in any of the baseline characteristics computed (age, sex, handedness, PCS, weekly distance span, ancillary use outside of home), was not found to explain variance, i.e., predicting value in PRWE score at 12 months. The R2 of the linear regression was 11.8% and the estimates of coefficients were for age 0.20 (CI: -0.67–1.07) and for PCS at baseline 0.16 (CI: -0.21–0.52).

## Discussion

In this randomized, multicenter, pragmatic trial, we compared FC and VFUDC positions in patients aged 65 years or more with dorsally displaced DRF. At 12 months, we found minor differences favoring the FC position in primary and secondary outcomes including complications.

PRWE score at 12 months showed between group difference of 4.9 points when the published MCID of the PRWE is 11 points. The 95% CI of difference in means includes greater than MCID size of advantage of FC and zero points of difference, hence the possibility of FC being distinctly better, or equality of the interventions, cannot be excluded. Further, many of the secondary outcomes showed a tendency to favor the FC position but included the similarity of difference of means in 95% CI. The rate of surgery due to loss of reduction, number of cast changes, and stiffness and pain in the wrist or hand reported at outpatient clinic at 3 months all had minor differences favoring the FC position.

In the present study, there were 14 cast changes in the FC group and 25 in the VFUDC group. Studies investigating cast immobilization methods have not routinely reported on cast changes, but studies have reported various rates of cast changes [26,28–30]. In our study, it was found that those patients who had one or more cast changes had worse PRWE outcomes at 12 months compared with patients who had no cast changes.

The secondary outcomes of grip strength, qDASH score and PRWE score at 3 months and qDASH score and VAS of pain at 12 months showed a small but constant difference between the studied groups. Grip strength in proportion to the controlled side at 3 months was 51% in the FC group and 45% in the VFUDC group, which is in line with two other RCTs on elderly DRF patients that reported grip strength in proportion to controlled side at 3 months to be 58% and 47% in non-operative groups [12,31].

In 1991, Gupta published the results of three different immobilization positions with 204 patients: volar-flexion, neutral, and dorsal-flexion, and found in favor of the dorsal-flexion group [6]. Van der Linden and Ericson studied 250 patients randomly assigned to five different immobilization positions and found that position had no importance regarding the final results [32]. Rajan et al. found better grip strength and less pain, disability, and limitation of movements in a dorsal-flexion group compared with a volar-flexion group [33]. Grle et al. studied 100 patients and found that dorsal-flexion was of minor benefit compared with volar-flexion at 2-month follow-up [34]. The results of our study, in line with the findings of the above studies, suggest that volar-flexion is not a superior cast position after dorsally displaced DRF.

Although some studies have reported the prognostic and explanatory value of various psychological factors on the PROM outcomes of DRF [17–19], this study did not find a prognostic value of PCS at baseline for PRWE measured at 12 months, as was hypothesized, though a modest correlation value of PCS at 3 months with PRWE at 12 months was present. However, the rise in the patient-specific value of PCS between baseline and 12 months did show moderate correlation in PRWE at 12 months, which suggests that pain avoidance and catastrophic thinking related to pain is dynamic in nature and evolves in association with new experiences.

The assessed mean values of the radiographical indicators of inclusion validity and procedural validity after the closed reduction indicate that the recruiting on call physicians enrolled patients who were well aligned to the Finnish Current Care Guidelines [24] used at the time of enrolment to determine between non-operative and operative treatment. In this respect, there were no considerable differences between the findings of the three participating centers in the study. Furthermore, the internal validity of the two cast positions were assessed by two radiographical indicators which showed mean differences in radial angulations by the two studied interventions, as expected.

The rates of reported complications were low in both groups. We found, however, that the VFUDC group reported more stiffness and pain at outpatient clinic at 3 months. When taken together with the biomechanical study [35], this finding might suggest an elevated risk of median nerve compression in the VFUDC position when compared with the FC position. Moreover, two patients in the VFUDC group and none in FC group underwent ENMG on their injured forearm during follow-up.

This study has several limitations that should be considered. The follow-up of patients aged 65 years or more with dorsally displaced DRF at the time of closed reduction was limited to 12 months. The cast immobilization methods studied were limited to two widely used cast positions, although several other immobilization methods have been presented in the literature [2–6]. The sample size used does not permit convincing conclusions for infrequent outcomes, such as the difference in rate of complications, nor for the small differences between groups in continuous variables between the two studied interventions. Moreover, the SD of our primary outcome, PRWE score at 12 months, was larger than anticipated and utilized in a priori power calculation.

## Conclusions

In summary, the authors suggest that FC immobilization might lead to slightly more beneficial subjective functional outcomes with fewer complications when compared with VFUDC immobilization in the treatment of this common fracture but the similarity of the outcomes in 95% confidence interval cannot be excluded. In the clinical context of DRF treatment, the arm has to be immobilized to some cast position and, taken together with the results of our and previous studies, FC is more likely to result in superior outcomes than VFUDC.

## Supporting information

**S1 Checklist. The CONSORT checklist.**
(DOC)

**S1 Appendix. Inclusion and exclusion criteria of the study.**
(DOCX)

**S2 Appendix. Description of changes to the published protocol.**
(DOCX)

**S3 Appendix. Additional analyses of the data.**
(DOCX)

**S4 Appendix. Pictures of the functional cast and volar-flexion ulnar deviation cast.**
(DOCX)

**S1 Data.**
(XLSX)

**S1 File.**
(DOC)

## Acknowledgments

Authors of NITEP-group:

The authors of NITEP group are Lauri Raittio, B.Sc., B.M., Antti P. Launonen, M.D., Ph.D., Teemu Hevonkorpi, B.M., Toni Luokkala, M.D., Juha Kukkonen, M.D., Ph.D., Aleksi Reito, M.D., Ph.D., Minna K. Laitinen, M.D., Ph.D., Ville M. Mattila, M.D., Ph.D.

## Author Contributions

**Conceptualization:** Antti P. Launonen, Teemu Hevonkorpi, Toni Luokkala, Juha Kukkonen, Aleksi Reito, Minna K. Laitinen, Ville M. Mattila.

**Data curation:** Lauri Raittio, Aleksi Reito, Ville M. Mattila.

**Formal analysis:** Lauri Raittio, Aleksi Reito.

**Funding acquisition:** Antti P. Launonen, Toni Luokkala, Juha Kukkonen, Aleksi Reito, Minna K. Laitinen, Ville M. Mattila.

**Investigation:** Lauri Raittio, Antti P. Launonen, Toni Luokkala, Juha Kukkonen, Aleksi Reito, Minna K. Laitinen, Ville M. Mattila.

**Methodology:** Lauri Raittio, Antti P. Launonen, Teemu Hevonkorpi, Toni Luokkala, Juha Kukkonen, Aleksi Reito, Minna K. Laitinen, Ville M. Mattila.

**Project administration:** Lauri Raittio, Antti P. Launonen, Teemu Hevonkorpi, Juha Kukkonen, Minna K. Laitinen, Ville M. Mattila.

**Resources:** Antti P. Launonen, Teemu Hevonkorpi, Juha Kukkonen, Minna K. Laitinen, Ville M. Mattila.

**Software:** Lauri Raittio, Antti P. Launonen, Ville M. Mattila.

**Supervision:** Antti P. Launonen, Teemu Hevonkorpi, Juha Kukkonen, Minna K. Laitinen, Ville M. Mattila.

**Validation:** Lauri Raittio, Antti P. Launonen, Juha Kukkonen, Aleksi Reito, Minna K. Laitinen, Ville M. Mattila.

**Writing – original draft:** Lauri Raittio, Teemu Hevonkorpi.

**Writing – review & editing:** Lauri Raittio, Antti P. Launonen, Teemu Hevonkorpi, Toni Luokkala, Juha Kukkonen, Aleksi Reito, Minna K. Laitinen, Ville M. Mattila.

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
