## [Decision Letter · Decision Letter 0]

27 Oct 2019

PONE-D-19-23143

Two casting methods compared in patients with Colles' fracture: A pragmatic, randomized controlled trial

PLOS ONE

Dear Mr Raittio,

Thank you for submitting your manuscript to PLOS ONE. After careful consideration, we feel that it has merit but does not fully meet PLOS ONE’s publication criteria as it currently stands. Therefore, we invite you to submit a revised version of the manuscript that addresses the points raised during the review process.

The manuscript has been assessed by two reviewers, their comments are available below.

The reviewers are positive about the study but they have requested some clarifications about aspects of the methodology and additional information on the analyses completed. Could you please revise the manuscript to address the items raised by the reviewers?

We would appreciate receiving your revised manuscript by Dec 09 2019 11:59PM. Please include the following items when submitting your revised manuscript:

We look forward to receiving your revised manuscript.

Kind regards,

Iratxe Puebla

Senior Managing Editor, PLOS ONE

Journal Requirements:

2. Thank you for submitting your clinical trial to PLOS ONE and for providing the name of the registry and the registration number. The information in the registry entry suggests that your trial was registered after patient recruitment began. PLOS ONE strongly encourages authors to register all trials before recruiting the first participant in a study.

a) your reasons for your delay in registering this study (after enrollment of participants started);

b) confirmation that all related trials are registered by stating: “The authors confirm that all ongoing and related trials for this drug/intervention are registered”.

Please also ensure you report the date at which the ethics committee approved the study as well as the complete date range for patient recruitment and follow-up in the Methods section of your manuscript.

3. One of the noted authors is a group or consortium NITEP group. In addition to naming the author group, please list the individual authors and affiliations within this group in the acknowledgments section of your manuscript. Please also indicate clearly a lead author for this group along with a contact email address.

Reviewers' comments:

Reviewer's Responses to Questions

**Comments to the Author**

1. Is the manuscript technically sound, and do the data support the conclusions?

Reviewer #1: Yes

Reviewer #2: Yes

2. Has the statistical analysis been performed appropriately and rigorously? 

Reviewer #1: Yes

Reviewer #2: Yes

3. Have the authors made all data underlying the findings in their manuscript fully available?

Reviewer #1: Yes

Reviewer #2: No

4. Is the manuscript presented in an intelligible fashion and written in standard English?

Reviewer #1: Yes

Reviewer #2: No

5. Review Comments to the Author

Reviewer #1: well done study statistical methodology sound

addresses a contemporary question now that so many of these fractures are treated operatively

This reviewer has little critique with your overall methodology nor conclusions

Reviewer #2: This investigation is well and completely reported and discussed. The English language is poor, even unclear, in parts (e.g., section ‘PCS’ in the appendix) and needs to be revised by a native speaker or other competent person. I have only minor comments and suggestions:

1. Please state the randomisation block size. According to the stated stratification factors, there seem to have been 2x3x2 = 12 strata, which for planned 57 patients per group would seem to lead to very numbers in each stratum.

2. The statistics software used for data analysis should be stated.

3. Detail of the multiple imputation procedure used should be given in the appendix.

4. Please state which outcome variables were collected at baseline, and the time-point of the outcome. In Table 2, only PCS is given at baseline. Further, which ‘difference’ is meant in the sentence beginning ‘The primary outcome measure was the difference…’, p.5? I understand that the PRWE was not measured at baseline, so presumably you mean the difference between the treatment groups? Generally, an ‘outcome’ should be defined on the individual patient.

5. I assumed that all localised outcome measures refer to the injured wrist, but on p.8 we are informed about the correlation with grip strength of the uninjured limb – please clarify.

6. When were the radiograph examinations and assessments performed, and were the assessors blinded to treatment group?

7. I suggest displaying the distribution of primary (and possibly important secondary) outcome values graphically, for instance using box-plots or scatter plots comparing the treatment groups.

8. Were subgroup analyses per site performed and if so, were any differences found? In such a surgery study, where physician-specific effects may be expected, it would be important to look for site subgroup differences in the primary treatment comparison.

9. On p.13 (discussion) a cast change rate per patient of 260% is referred to – does this mean multiple changes in some patients?

6. PLOS authors have the option to publish the peer review history of their article (what does this mean?). If published, this will include your full peer review and any attached files.

Reviewer #1: No

Reviewer #2: Yes: Jeremy Franklin

---

## [Author Response · Author response to Decision Letter 0]

7 Nov 2019

Rebuttal letter

We would like to take this opportunity to thank the Reviewers for their constructive criticism of our manuscript. We have strived to revise the manuscript in accordance with the valuable criticism received from the Reviewers. We feel that the quality of our manuscript has improved as a result. 

The comments of the Reviewers are followed by our answers and a description of the corrections/changes that we have made to the manuscript (marked in italics).

Reviewers' comments:

1. Is the manuscript technically sound, and do the data support the conclusions?

Reviewer #1: Yes

Reviewer #2: Yes

2. Has the statistical analysis been performed appropriately and rigorously?

Reviewer #1: Yes

Reviewer #2: Yes

3. Have the authors made all data underlying the findings in their manuscript fully available?

Reviewer #1: Yes

Reviewer #2: No

We have now verified the policies of our Regional Ethical Committee on data sharing. In order to ensure the anonymization of the dat. Hence, the exact day and month of the recruitment has been removed and study centers are only referred to as numbers one, two, and three. Due to the common nature of distal radius fracture, we feel these changes enable the required anonymization of the data. The data will be submitted in the supplementary file alongside the manuscript. 

4. Is the manuscript presented in an intelligible fashion and written in standard English?

Reviewer #1: Yes

Reviewer #2: No

5. Review Comments to the Author

Reviewer #1: well done study statistical methodology sound addresses a contemporary question now that so many of these fractures are treated operatively. This reviewer has little critique with your overall methodology nor conclusions.

The authors thank the reviewer for the positive comments.

Reviewer #2: This investigation is well and completely reported and discussed. The English language is poor, even unclear, in parts (e.g., section ‘PCS’ in the appendix) and needs to be revised by a native speaker or other competent person. I have only minor comments and suggestions:

The authors thank the reviewer for the positive comments. The material in the supplementary files has now been revised by a native speaker of English. 

1. Please state the randomisation block size. According to the stated stratification factors, there seem to have been 2x3x2 = 12 strata, which for planned 57 patients per group would seem to lead to very numbers in each stratum.

In the randomization process, we had two stratification variables, age and extra- or intra-articular fracture line, with two blocks in each variable. Thus, there were 4 stratification factors (lines 116-7).

2. The statistics software used for data analysis should be stated.

 The statistical software used for the data analysis was SPSS v. 25 (lines 147-8).

3. Detail of the multiple imputation procedure used should be given in the appendix.

The variables used for the multiple imputation were age, sex, cast position (treatment allocation), the PRWE at 3 and 12 months, the PCS at 3 and 12 months, the VAS for pain at 3 and 12 months, and the Quick-DASH at 3 and 12 months. This has now been explained in more detail in appendix S4.

4. Please state which outcome variables were collected at baseline, and the time-point of the outcome. In Table 2, only PCS is given at baseline. Further, which ‘difference’ is meant in the sentence beginning ‘The primary outcome measure was the difference…’, p.5? I understand that the PRWE was not measured at baseline, so presumably you mean the difference between the treatment groups? Generally, an ‘outcome’ should be defined on the individual patient.

The mean difference between study groups at 12 months was the primary outcome. This has now been stated more clearly (lines 126-7).

5. I assumed that all localised outcome measures refer to the injured wrist, but on p.8 we are informed about the correlation with grip strength of the uninjured limb – please clarify.

The reviewer is indeed correct that we have reported the correlation with the grip strength of the uninjured limb to the PRWE score in order to find an association, if any, between the physical strength of the upper arms and the primary outcome. 

6. When were the radiograph examinations and assessments performed, and were the assessors blinded to treatment group?

The radiograph examinations were performed before and after the closed reduction, 1, 2, and 5 weeks after fracture. The radiographic assessments were performed by authors who were blinded to the treatment allocation at the time of assessments (lines 133-4).

7. I suggest displaying the distribution of primary (and possibly important secondary) outcome values graphically, for instance using box-plots or scatter plots comparing the treatment groups.

The authors fully agree with the usefulness of the graphical reporting of the main results. We have now added a box-plot graph that contains the results of the PRWE, the Quick-DASH, and VAS for pain in appendix S4.

8. Were subgroup analyses per site performed and if so, were any differences found? In such a surgery study, where physician-specific effects may be expected, it would be important to look for site subgroup differences in the primary treatment comparison.

In each study center, recruitment to the study and the closed reduction of the fracture was performed by various on-call physicians in the emergency departments. Therefore, each physician only recruited and treated a few patients in the study. The subgroup analysis of radiograph parameters for the internal validation of the allocated treatment showed no difference between study centers. 

9. On p.13 (discussion) a cast change rate per patient of 260% is referred to – does this mean multiple changes in some patients?

This rate of cast changes has been referred to in a paper published by Anzarut et a.l (ref 26 in the manuscript) who reported in Table 1 of the article the mean of 2.6 and the median of 2 cast changes in the study population. However, we are not confident in this reported number and we have therefore changed the sentence: “Studies investigating cast immobilization methods have not routinely reported on cast changes, but studies have reported various rates of cast changes” (lines 270-1).

---

## [Decision Letter · Decision Letter 1]

9 Apr 2020

Two casting methods compared in patients with Colles' fracture: A pragmatic, randomized controlled trial

PONE-D-19-23143R1

Dear Dr. Raittio,

We are pleased to inform you that your manuscript has been judged scientifically suitable for publication and will be formally accepted for publication once it complies with all outstanding technical requirements.

With kind regards,

Jesse Bernard Jupiter, MD

Guest Editor

PLOS ONE

Additional Editor Comments (optional):

Reviewers' comments:

Reviewer's Responses to Questions

**Comments to the Author**

1. If the authors have adequately addressed your comments raised in a previous round of review and you feel that this manuscript is now acceptable for publication, you may indicate that here to bypass the “Comments to the Author” section, enter your conflict of interest statement in the “Confidential to Editor” section, and submit your "Accept" recommendation.

Reviewer #1: All comments have been addressed

Reviewer #2: All comments have been addressed

2. Is the manuscript technically sound, and do the data support the conclusions?

Reviewer #1: Yes

Reviewer #2: (No Response)

3. Has the statistical analysis been performed appropriately and rigorously? 

Reviewer #1: Yes

Reviewer #2: (No Response)

4. Have the authors made all data underlying the findings in their manuscript fully available?

Reviewer #1: Yes

Reviewer #2: (No Response)

5. Is the manuscript presented in an intelligible fashion and written in standard English?

Reviewer #1: Yes

Reviewer #2: (No Response)

6. Review Comments to the Author

Reviewer #1: no further comments. The authors have fully complied with the suggests made by the reviewers and improved the overall clarity

Reviewer #2: (No Response)

7. PLOS authors have the option to publish the peer review history of their article (what does this mean?). If published, this will include your full peer review and any attached files.

Reviewer #1: No

Reviewer #2: Yes: Jeremy Franklin

---

## [Editor Report · Acceptance letter]

27 Apr 2020

PONE-D-19-23143R1 

Two casting methods compared in patients with Colles' fracture: A pragmatic, randomized controlled trial 

Dear Dr. Raittio:

I am pleased to inform you that your manuscript has been deemed suitable for publication in PLOS ONE. Congratulations! Your manuscript is now with our production department. 

With kind regards,

on behalf of

Dr. PLOS Manuscript Reassignment 

Staff Editor

PLOS ONE